# Sperm-Cultured Gate Ion-Sensitive Field-Effect Transistor for Non-Optical and Live Monitoring of Sperm Capacitation

**DOI:** 10.3390/s19081784

**Published:** 2019-04-14

**Authors:** Akiko Saito, Toshiya Sakata

**Affiliations:** Department of Materials Engineering, School of Engineering, The University of Tokyo, Tokyo 113-8656, Japan; saito@biofet.t.u-tokyo.ac.jp

**Keywords:** artificially induced sperm capacitation, field-effect transistor (FET), pH, progesterone, calcium ion

## Abstract

We have successfully monitored the effect of progesterone and Ca^2+^ on artificially induced sperm capacitation in a real-time, noninvasive and label-free manner using an ion-sensitive field-effect transistor (ISFET) sensor. The sperm activity can be electrically detected as a change in pH generated by sperm respiration based on the principle of the ISFET sensor. Upon adding mouse sperm to the gate of the ISFET sensor in the culture medium with progesterone, the pH decreases with an increasing concentration of progesterone from 1 to 40 μM. This is because progesterone induces Ca^2+^ influx into spermatozoa and triggers multiple Ca^2+^-dependent physiological responses, which subsequently activates sperm respiration. Moreover, this pH response of the ISFET sensor is not observed for a Ca^2+^-free medium even when progesterone is introduced, which means that Ca^2+^ influx is necessary for sperm activation that results in sperm capacitation. Thus, a platform based on the ISFET sensor system can provide a simple method of evaluating artificially induced sperm capacitation in the field of male infertility treatment.

## 1. Introduction

Sperm capacitation is necessary for successful fertilization following sperm hyperactivation, acrosome reaction and chemotaxis towards the egg [1,2,3,4,5]. These events are recognized as Ca^2+^-dependent physiological responses. Moreover, the steroid hormone progesterone, which is released by the cumulus cells surrounding the egg, contributes to Ca^2+^ influx into spermatozoa and triggers multiple Ca^2+^-dependent physiological responses [6,7,8]. Microscopic observations are the mainstream [9,10,11] methods for evaluating sperm behaviors, such as acrosome reaction and chemotaxis of spermatozoa, but optical setups are mostly expensive and the preparation of reagents is time-consuming. The patch-clamp technique has also used to detect Ca^2+^ influx into spermatozoa [7,12], but a fine glass pipette tip has to be carefully pressed against the sperm membrane and a tiny area of the sperm membrane is sucked into it, which is basically an invasive and tedious method. Therefore, it would be highly beneficial to develop a non-optical and noninvasive method for the quantitative evaluation of sperm behaviors. In addition, intracellular Ca^2+^ regulates various cellular functions, including mitochondrial metabolism [13,14] and thus, sperm respiration may be available as an indicator of sperm capacitation.

As one of the detection principles to analyze cellular behaviors, a cell-cultured gate ion-sensitive field-effect transistor (ISFET) is used to detect them quantitatively as a change in pH during cell culture in a non-optical and noninvasive manner [15,16,17,18]. Because the gate insulating membrane usually consists of an oxide with hydroxy groups at the surface in a solution, ISFET sensors are responsive to the change in the concentration of positively charged hydrogen ions based on the equilibrium reaction. Thus, they are used as pH sensors [19,20]. In fact, cellular respiration was monitored as the change in pH based on the amount of respiration products (CO_2_ or lactic acid) released from cells. Furthermore, the change in pH at the cell/gate nanogap interface was found to be closely related to the quantitative electrical signal for adhesive cells using the gate electrode of an ISFET sensor [17]. Therefore, the cell-coupled gate ISFET sensor may allow for the quantitative monitoring of sperm activities based on sperm metabolism in a non-optical and noninvasive manner.

Mammalian sperm may obtain access to their conspecific egg by metabolic energy production in the form of ATP, following sperm capacitation [21]. The metabolic energy production may be fueled by the Embden–Meyerhof pathway of glycolysis, mitochondrial oxidative phosphorylation or a combination of both pathways [22,23,24]. As a result, the respiration products, such as CO_2_ and lactic acid, are expected to contribute to the change in pH around spermatozoa, that is, such sperm respiration can be detected using a cell-based ISFET sensor. This means that the change in pH during sperm culture may potentially be an indicator to estimate sperm capacitation.

In this paper, we propose a sperm-cultured gate ISFET sensor for the continuous monitoring of sperm respiration. In particular, the effect of progesterone and Ca^2+^ on artificially induced sperm capacitation is investigated in a non-optical and noninvasive manner.

## 2. Materials and Methods

### 2.1. Preparation of Mouse Sperm

All the nutrient media used for mouse sperm were prepared the day before the experiments and the balance of gases in the droplet of sperm was controlled overnight under suitable conditions (5% CO_2_ and 37 °C). As a culture medium, human tubal fluid (HTF) was prepared, which included NaCl (102 mM; Wako), KCl (4.7 mM; Wako., Japan), MgSO_4_·7H_2_O (0.2 mM; Wako), KH_2_PO_4_ (0.4 mM; Wako), CaCl_2_·2H_2_O (2.0 mM; Wako), NaHCO_3_ (25 mM; Wako), glucose (2.8 mM; Wako), Na lactate (23.2 mM; Wako), Na pyruvate (0.3 mM; Wako), 0.3% BSA (Sigma., Kanagawa, Japan), penicillin G (100 U/ml; Meiji., Tokyo, Japan) and streptomycin (100 µg/ml; Meiji). Male ICR mice (9−10 weeks old; Charles River Japan Inc., Yokohama-shi, Japan) were sacrificed by cervical dislocation. Spermatozoa collected from the cauda epididymides of ICR male mice were incubated for 1.5 h at 5% CO_2_ and 37 °C. In this case, 1 × 10^5^ or 1 × 10^6^ cells/mL of spermatozoa were seeded in an HTF droplet of 200 μL before being cultured again in fresh HTF or Ca^2+^-free HTF (CaCl_2_·2H_2_O was removed from the prepared HTF) at 5% CO_2_ and 37 °C. In general, the number of spermatozoa that can be collected from a male mouse is known to be about 1 × 10^6^ /mL or 5 × 10^6^ /mL at most. We obtained Institutional Review Board approval for this study. This study was overseen by an Animal Care and Use Committee at The University of Tokyo.

### 2.2. Electrical Measurement with ISFET Sensor

We used a n-channel depletion-mode FET (silicon) with a Ta_2_O_5_/SiO_2_ (100 nm/50 nm) layer as a gate insulator, which had a width (*W*) and length (*L*) of 340 and 10 μm, respectively (ISFETCOM Co., Ltd., Saitama, Japan), as shown in Figure 1a. To prevent leakage currents in a buffer solution, the Ta_2_O_5_ thin film was coated and used as a passivation layer on the gate. The gate voltage (*V*_G_)–drain current (*I*_D_) electrical characteristics were measured using a semiconductor parameter analyzer (B1500A, Agilent, Hachioji-shi, Japan). The change in *V*_G_ in the *V*_G_–*I*_D_ electrical characteristics was estimated as the shift in the threshold voltage (*V*_T_), which was evaluated at a constant *I*_D_ of 1 mA and a constant of drain voltage (*V*_D_) of 2 V. The surface potential at the gate surface (*V*_out_) was continuously monitored using a source follower circuit (Appendix A), with which the potential change at the interface between a gate insulator and an aqueous solution can be directly output at a constant *I*_D_ (RadianceWare Inc., Saitama, Japan) when *V*_D_ and *I*_D_ were set to 2.5 V and 700 μA, respectively. pH standard buffer solutions (pH 4.01, 6.86, 7.41 and 9.18; Wako) were prepared as the measurement solutions for the analysis of pH sensitivity with the ISFET sensors and a culture medium was utilized for the measurement of sperm behaviors. An Ag/AgCl reference electrode with KCl solution (Wako) was connected to the measurement solution through a salt bridge. 

In the electrical measurement of the sperm-cultured gate ISFET sensor, the HTF culture medium with or without progesterone (Wako) was utilized as a measurement solution at 5% CO_2_ and 37 °C. The concentration of progesterone was changed from 1 to 60 μM in the culture medium after it was dissolved in dimethyl sulfoxide (DMSO; Sigma, Kanagawa, Japan) at a concentration of 1 mM. The Ca^2+^-free HTF medium with progesterone (1 to 60 μM) and Ca^2+^ ionophore (10 μM; Sigma) was also utilized to investigate the effect of Ca^2+^ on the electrical signal. Mouse spermatozoa, which were pretreated in HTF or Ca^2+^-free HTF, were introduced at 0 h in all the experiments using the sperm-cultured gate ISFET sensors. The ISFETs used in this study showed a negligible drift signal of less than 5 mV for a long-time measurement (about 100 h) in the culture medium.

## 3. Results

### 3.1. Concept of Sperm-Cultured Gate ISFET Sensor

Figure 1a shows the concept of the sperm-cultured gate ISFET sensor. Spermatozoa were cultured at the concentration of 1 × 10^5^ or 1 × 10^6^ /mL on the gate (Ta_2_O_5_/SiO_2_) of the ISFET in the polycarbonate ring with a diameter of 18 mm (1 mL) where they were actively moving. That is, the change in pH in the polycarbonate ring was continuously monitored using the ISFET sensor. The semiconducting material is separated from the solution by the gate insulator, the thickness of which is in the order of 100 nm. The gate insulator is often composed of an oxide, such as SiO_2_, Ta_2_O_5_, Al_2_O_3_ or nitrides (e.g., Si_3_N_4_) [25]. The hydroxy groups are formed at the surface of the gate insulator in the solution and are thus responsive to hydrogen ions (Figure 1a). These positive charges at the gate surface electrostatically interact with electrons at the channel in the silicon substrate. The field effect induced by the changes in the charge densities at the gate causes a change in the threshold voltage (Δ*V*_T_) at a constant drain source current (*I*_D_) in the gate voltage (*V*_G_)–*I*_D_ electrical characteristic. This electrical response of the ISFET to hydrogen ions is Nernstian at about 59.1 mV/pH at 25 °C [19,20]. In fact, Δ*V*_T_ was measured with changes in pH using one of the ISFET sensors in this study (Figure 1b). From such electrical characteristics, the average pH sensitivity was found to be about 56 mV/pH at 25 °C for the 15 ISFET sensors used in this study, as shown in Figure 1c. According to the detection principle, the surface potential at the gate surface was continuously monitored using a source follower circuit. As a result, the potential change (Δ*V*_out_) at the interface between the gate insulator and an aqueous solution, which corresponds to -Δ*V*_T_, can be directly output at a constant *I*_D_ [26]. Three variables, namely Δ*V*_out_ at the gate with a changing pH, pNa^+^ and pK^+^, are shown in Figure 2. The pH sensitivity was about 56 mV/pH, which was consistent with that obtained from the measurement of Δ*V*_T_, while no electrical response was found for the change in Na^+^ concentration (Figure 2a) or K^+^ concentration (Figure 2b). Thus, the pH sensitivity and selectivity for the original ISFET sensors were quantitatively estimated from the calibration curve and these sensors were utilized as the sperm-cultured gate ISFET sensors in this study.

### 3.2. Electrical Monitoring of Sperm Respiration with Sperm-Cultured Gate ISFET Sensor: Effect of Progesterone and Ca^2+^

Figure 3a shows Δ*V*_out_ that was measured using the sperm-cultured gate ISFET sensor with the concentration of spermatozoa being 1 × 10^5^ /mL. In this case, progesterone was included to stimulate spermatozoa in the culture medium at a concentration of 1–60 μM. However, no electrical signal was found even after adding progesterone for the case of 1 × 10^5^ /mL spermatozoa. On the other hand, Figure 3b shows Δ*V*_out_ that was measured using the sperm-cultured gate ISFET sensor with the concentration of spermatozoa being 1 × 10^6^ /mL. Δ*V*_out_ for the sperm-cultured gate ISFET sensors increased with the time after the introduction of spermatozoa. In particular, Δ*V*_out_ for the sperm-cultured gate ISFET sensors increased when the concentration of progesterone was increased to 40 μM although Δ*V*_out_ appeared to be saturated at 60 μM. Moreover, the effect of Ca^2+^ on sperm activities was investigated using the sperm-cultured gate ISFET sensor. That is, the Ca^2+^-free culture medium was used for the electrical measurement. After this, the Ca ionophore was added to the medium to induce the acrosome reaction [27]. As shown in Figure 3c, Δ*V*_out_ for the sperm-cultured gate ISFET sensor hardly changed after adding progesterone to the Ca^2+^-free culture medium, even when 1 × 10^6^/mL of spermatozoa were cultured and Ca ionophore was included. On the other hand, Δ*V*_out_ increased to more than 50 mV upon the addition of Ca ionophore to the sperm-cultured gate ISFET sensor in the HTF medium with Ca^2+^ and progesterone (40 μM), as shown in Figure 4.

## 4. Discussion

The concentration of spermatozoa in a measurement solution should be controlled to obtain the electrical signal of the sperm-cultured gate ISFET sensor. In fact, hardly any electrical response was found for 1×10^5^ /mL of spermatozoa. Furthermore, 1 × 10^6^ /mL of spermatozoa were required for the sperm-cultured gate ISFET sensor to evaluate sperm activities (Figure 3). The positive shift of Δ*V*_out_ obtained in Figure 3b indicated an increase in positive charges at the gate surface. That is, the concentration of hydrogen ions increased based on sperm respiration due to the high sensitivity and selectivity of ISFET sensors to pH (Figure 1). As mentioned in the Introduction section, spermatozoa are activated to reach and fertilize an egg, which is accompanied by the production of ATP energy based on sperm metabolism. Consequently, the release of CO_2_ or lactic acid from spermatozoa causes the acidification of the measurement solution when the metabolic pathway is glycolysis or oxidative phosphorylation. Moreover, the addition of progesterone contributed to an increase in Δ*V*_out_, which ultimately contributed to the activation of sperm respiration (Figure 3). In particular, the limitation of progesterone concentration in terms of its effect was found to be 40 μM for 1 × 10^6^ /mL of spermatozoa. This limitation is expected to depend on the number of spermatozoa in the measurement solution. In fact, it is well known that progesterone contributes to Ca^2+^ influx into spermatozoa and triggers the multiple Ca^2+^-dependent physiological responses that result in successful fertilization. Therefore, the increase in Δ*V*_out_ after the addition of progesterone indicates the enhancement of sperm activities, such as sperm respiration. This consideration is also confirmed by the results in Figure 3c. A Ca^2+^-free medium did not induce Ca^2+^ influx into spermatozoa even after adding progesterone, which resulted in no induction of sperm respiration, although sperm respiration in itself should normally occur because glucose was included in the culture medium. Moreover, the HTF medium including both progesterone and Ca ionophore must have greatly enhanced Ca^2+^ influx into spermatozoa, which contributed to a change in the electrical signal that resulted from sperm respiration (Figure 4). Thus, the sperm-cultured gate ISFET sensor is suitable for the non-optical and live monitoring of sperm metabolism, enabling the evaluation of sperm capacitation. In the future, a platform with the sperm-cultured gate ISFET sensor should be combined with a microfluidic device as a miniaturized system for male infertility diagnostics since the ISFET chip can be miniaturized by a conventional semiconductor process.

Considering the average pH sensitivity (56 mV/pH) of the ISFETs shown in Figure 1c, the change in pH for the sperm-cultured gate ISFET sensors can be calculated. Figure 5 shows the change in pH for the sperm-cultured gate ISFET sensors that corresponds to each result plotted in Figure 3. The pH of the original medium was 7.4. When 40 μM progesterone was included in the culture medium, the change in pH reached about 0.2 at a time of 25 min after the introduction of spermatozoa. This means that progesterone induced a large shift in pH due to the activation of sperm respiration even in the buffered medium. In particular, the enhancement of Ca^2+^ influx with Ca ionophore increased the change in pH (about 0.9) based on the potential response of the sperm-cultured gate ISFET sensor (Figure 4). In fact, intracellular Ca^2+^ regulates numerous cellular functions, including mitochondrial metabolism, muscle contraction, cellular motility and vesicular transport, through the function of calcium-binding proteins [13,14]. Therefore, Ca^2+^ influx into spermatozoa based on progesterone stimulation should have induced sperm respiration, which contributed to the change in pH obtained by the sperm-cultured gate ISFET sensor. However, it could not be determined which metabolic pathway was preferred, namely glycolysis or oxidative phosphorylation, from the change in pH based on the electrical signal of the ISFET sensor. If oxidative phosphorylation is preferred for the sperm metabolism, the amounts of ATP and CO_2_ can be roughly estimated for the cultured spermatozoa based on the change in pH. According to the scheme shown in Appendix A, the concentrations of released CO_2_ and generated ATP for 1 × 10^6^ /mL of spermatozoa in the glass ring (1 mL) were calculated to be about 3% (1.2 mM) and about 7.5 mM, respectively. That is, about 7.5 pmol of ATP was generated per spermatozoon. In fact, the ATP content was reported to be 100–300 amol per spermatozoon in a previous study [28]. Therefore, the change in ATP content calculated in this study appeared to be relatively high. This is assumed to be because the change in pH with the sperm-cultured gate ISFET sensor was caused by not only the dissolution of CO_2_ based on oxidative phosphorylation but also the release of lactic acid based on anaerobic glycolysis [21]. However, the change in the ATP concentration in spermatozoa can also be caused by other mechanisms. Adenylate cyclase has greater levels of activation when accompanied by sperm capacitation, resulting in the generation of cAMP by the consumption of ATP [29]. This is why more ATP may be produced by metabolic activities. In any case, it is clear that the progesterone stimulation of spermatozoa induced the decrease in extracellular pH.

## 5. Conclusions

In summary, we have demonstrated the non-optical and live monitoring of sperm capacitation using a sperm-cultured gate ISFET sensor. The ISFET sensor can directly detect changes in pH in a label-free and real-time manner and the change in pH based on sperm respiration was monitored using the sperm-cultured gate ISFET sensor. In particular, the stimulation of spermatozoa by progesterone, which triggers multiple Ca^2+^-dependent physiological responses, induced a change in pH of the sperm-cultured gate ISFET sensor, while no change was observed for a Ca^2+^-free culture medium even when Ca ionophore was added. Based on previous works, progesterone was effective in inducing Ca^2+^ influx into spermatozoa, which resulted in the activation of mitochondrial metabolism. That is, sperm respiration is considered to have been enhanced by the progesterone stimulation. Thus, a platform based on the sperm-cultured gate ISFET sensor is suitable for the non-optical and noninvasive evaluation of sperm activities, such as capacitation, in the field of male infertility. In particular, the sperm-cultured gate ISFET sensor is expected to be combined in a microfluidic system in the future.

To detect sperm respiration as a change in pH using the sperm-cultured gate ISFET sensor, the spermatozoa need to be prepared in the chamber with the gate at a concentration of more than 1 × 10^6^ /mL, according to the data shown in Figure 3. This means that a sufficient number of spermatozoa are required for the evaluation by the induction of a pH change based on sperm respiration. The number of spermatozoa can be reduced by the use of a microfluidic system, while maintaining the concentration of spermatozoa. However, this limited number of spermatozoa may not be a problem for semen examination in male infertility because at least 1.5 × 10^7^ /mL spermatozoa are required, while azoospermia is suspected in the case of low sperm numbers [30]. Therefore, it is expected that this device will be used for human samples in the future.

## Figures and Tables

**Figure 1 sensors-19-01784-f001:**
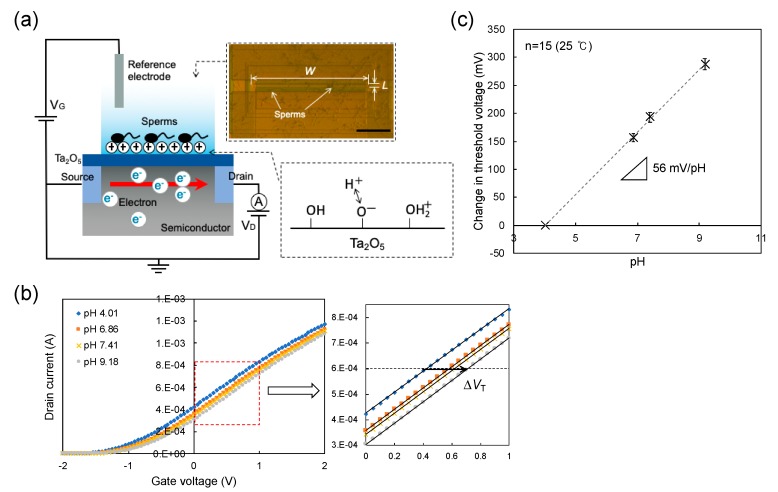
(**a**) Conceptual structure of sperm-cultured gate ISFET sensor for electrical measurements. The channel was designed to have a width (*W*) and length (*L*) of 340 and 10 μm, respectively. Scale bar = 100 μm. Hydroxy groups at the oxide membrane in a solution exhibit the equilibrium reaction with hydrogen ions. (**b**) *V*_G_–*I*_D_ electrical characteristic of one of the ISFET sensors used in this study. The shift in *V*_G_ at a constant *I*_D_ of 1 mA was estimated as the change in *V*_T_ when the pH was changed from a pH of 4.01 to 9.18. (**c**) Calibration curve, which was analyzed based on the data shown in (b). The pH sensitivity of this ISFET sensor was about 56 mV/pH, which almost showed a Nernstian response at 25 °C. *V*_G_ at a pH of 4.01 was offset to 0.

**Figure 2 sensors-19-01784-f002:**
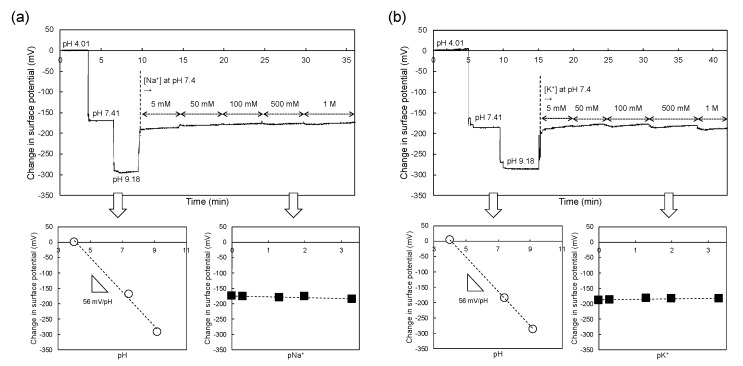
(**a**) Change in surface potential with varying pH and Na^+^ concentration using ISFET sensor. The gate surface potential at a pH of 4.01 was offset to 0 and the change in the potential was measured from a pH of 4.01 to 9.18 and then from a Na^+^ concentration of 5 mM to 1 M. (**b**) Change in surface potential with varying pH and K^+^ concentration using ISFET sensor. The gate surface potential at a pH of 4.01 was offset to 0 and the change in the potential was measured from a pH of 4.01 to 9.18 and then from a K^+^ concentration of 5 mM to 1 M.

**Figure 3 sensors-19-01784-f003:**
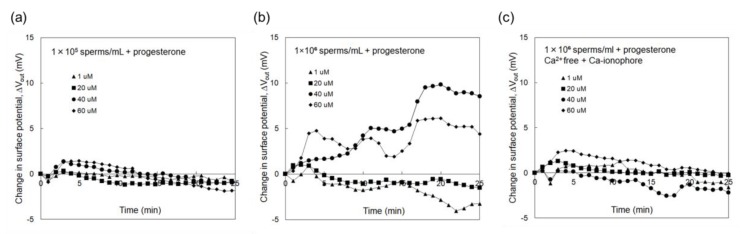
Change in surface potential of sperm-cultured gate ISFET sensor. Spermatozoa were added in the measurement medium at 0 h, which included progesterone at a concentration of 1–60 μM; (**a**) 1 × 10^5^ sperm/mL, (**b**) 1 × 10^6^ sperm/mL, (**c**) 1 × 10^6^ sperm/mL in the Ca^2+^-free HTF medium containing Ca ionophore.

**Figure 4 sensors-19-01784-f004:**
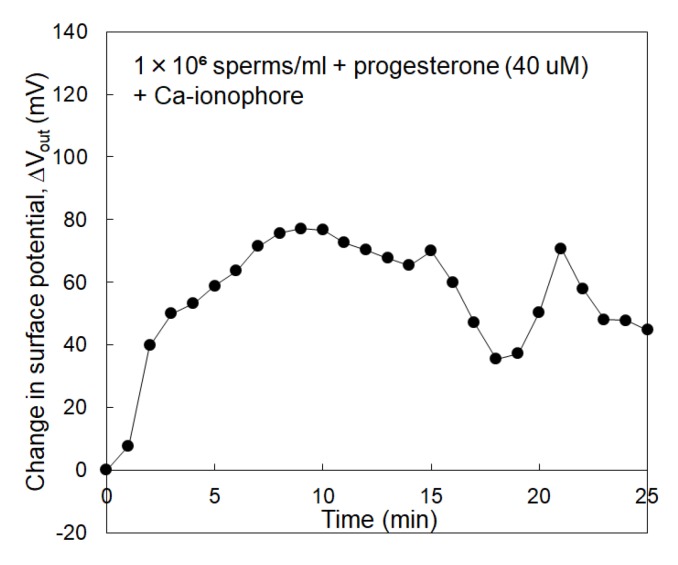
Change in surface potential of sperm-cultured gate ISFET sensor. A total of 1 × 10^6^ /mL of spermatozoa were added to the measurement medium at 0 h, which included 40 μM of progesterone and 10 μM of Ca ionophore.

**Figure 5 sensors-19-01784-f005:**
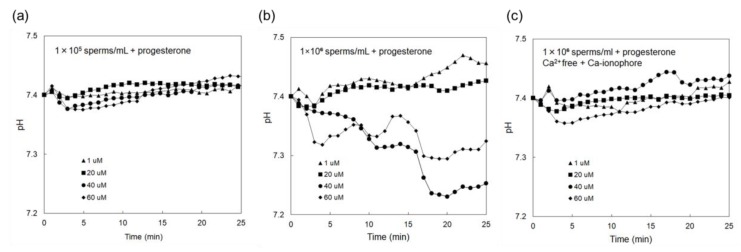
Change in pH of sperm-cultured gate ISFET sensor. Considering the pH sensitivity of the ISFET sensors (about 56 mV/pH on average) used in this study, the change in pH was calculated from the results plotted in Figure 3; (**a**) 1 × 10^5^ sperm/mL, (**b**) 1 × 10^6^ sperm/mL, (**c**) 1 × 10^6^ sperm/mL in Ca^2+^-free HTF medium containing Ca ionophore.

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
