# Peer review of "Sperm-Cultured Gate Ion-Sensitive Field-Effect Transistor for Non-Optical and Live Monitoring of Sperm Capacitation"

_sensors, 2019, doi:10.3390/s19081784_

Reviewer 1 Report

This work presents the effect of progesterone and Ca2+ on artificially induced sperm capacitation in a real-time, noninvasive, and label-free manner using an ion-sensitive field-effect transistor (ISFET) sensor. In general, it is an interesting work that can be used to screen the sperm activities. It can be accepted with minor suggestions and comments.

As the sperm cultured on the ISFET, there should be some variations of the ISFET read out because of the normal metabolism of the sperm and/or attachment of the sperm. It is suggested to demonstrate the time course of the surface potential change with sperm concentration larger than 10^6. It could also show the capability of the developed ISFET snesors.

It is widely known that there is long-time drifting of ISFET measurement within buffers. However, the drifting effect is not clearly observed with the experimental data in this work. I am wondering if there is any data process. If so, please describe the process in the manuscript that could help readers to understand the details.

Author Response

Dear Editor:

Thank you very much for your kind reviews. We have revised our manuscript in accordance with the reviewers’ comments as follows. We would be grateful if the manuscript could be revaluated for publication in Sensors.

[Reviewer: 1]

Comments and Suggestions for Authors:

This work presents the effect of progesterone and Ca2+ on artificially induced sperm capacitation in a real-time, noninvasive, and label-free manner using an ion-sensitive field-effect transistor (ISFET) sensor. In general, it is an interesting work that can be used to screen the sperm activities. It can be accepted with minor suggestions and comments.

As the sperm cultured on the ISFET, there should be some variations of the ISFET read out because of the normal metabolism of the sperm and/or attachment of the sperm. It is suggested to demonstrate the time course of the surface potential change with sperm concentration larger than 10^6. It could also show the capability of the developed ISFET sensors.

(Reply)

Thank you very much for the kind review. In this study, spermatozoa were collected from male mice for the continuous monitoring of sperm respiration. In general, the number of spermatozoa that can be collected from a male mouse is known to be about 1×106 /mL or 5×106 /mL at most. This is why we cannot definitely collect and use spermatozoa at the concentrations larger than 1×106 /mL. Therefore, we have added the following sentence in the revised manuscript.

“In general, the number of spermatozoa that can be collected from a male mouse is known to be about 1×106 /mL or 5×106 /mL at most.” (Line 73)

It is widely known that there is long-time drifting of ISFET measurement within buffers. However, the drifting effect is not clearly observed with the experimental data in this work. I am wondering if there is any data process. If so, please describe the process in the manuscript that could help readers to understand the details.

(Reply)

Thank you very much for the kind review. In fact, the ISFETs used in this study showed a negligible drift signal less than 5 mV for a long-time measurement (about 100 h) in the culture medium, which means that the drift effect was quite small for the measurement of sperm respiration (25 min). Therefore, we have added the following sentence in the revised manuscript.

“The ISFETs used in this study showed a negligible drift signal less than 5 mV for a long-time measurement (about 100 h) in the culture medium.” (Line 100)

Reviewer 2 Report

I review the paper by Saito and Sakata and I found it very interesting. The manuscript is well organized, and the scientific soundness of the presented result is valuable. Although the employment of ISFET devices for cell monitoring is not new, its application in the specific analyses described is quite innovative and can be effectively employed in real application scenario. Therefore, I suggest to accept the paper. I just have to highlight a small typo I found on row 237 ("nonoptocal", it should be "nonoptical"), so I suggest authors to revise their paper to correct other possible mistakes.

Author Response

Dear Editor:

Thank you very much for your kind reviews. We have revised our manuscript in accordance with the reviewers’ comments as follows. We would be grateful if the manuscript could be revaluated for publication in Sensors.

[Reviewer: 2]

Comments and Suggestions for Authors:

I review the paper by Saito and Sakata and I found it very interesting. The manuscript is well organized, and the scientific soundness of the presented result is valuable. Although the employment of ISFET devices for cell monitoring is not new, its application in the specific analyses described is quite innovative and can be effectively employed in real application scenario. Therefore, I suggest to accept the paper.

I just have to highlight a small typo I found on row 237 ("nonoptocal", it should be "nonoptical"), so I suggest authors to revise their paper to correct other possible mistakes.

(Reply)

Thank you very much for the kind review. We have checked and modified some typos in the revised manuscript.
